# Quality of Surface Texture and Mechanical Properties of PLA and PA-Based Material Reinforced with Carbon Fibers Manufactured by FDM and CFF 3D Printing Technologies

**DOI:** 10.3390/polym13111671

**Published:** 2021-05-21

**Authors:** Mohd Shahneel Saharudin, Jiri Hajnys, Tomasz Kozior, Damian Gogolewski, Paweł Zmarzły

**Affiliations:** 1Composites & Simulation Centre (CSC), Universiti Kuala Lumpur Malaysia Italy Design Institute (UniKL MIDI), Kuala Lumpur 56100, Malaysia; mshahneel@unikl.edu.my; 2Center of 3D Printing Protolab, Department of Machining, Assembly and Engineering Technology, Faculty of Mechanical Engineering, VSB Technical University Ostrava, 17. Listopadu 2172/15, 708 00 Ostrava-Poruba, Czech Republic; 3Department of Manufacturing Technology and Metrology, Kielce University of Technology, 25-314 Kielce, Poland; tkozior@tu.kielce.pl (T.K.); dgogolewski@tu.kielce.pl (D.G.); pzmarzly@tu.kielce.pl (P.Z.)

**Keywords:** 3D printing, carbon fibers, polymers, FDM, CFF, mechanical properties

## Abstract

The paper presents the results of mechanical tests of models manufactured with two 3D printing technologies, FDM and CFF. Both technologies use PLA or PA-based materials reinforced with carbon fibers. The work includes both uniaxial tensile tests of the tested materials and metrological measurements of surfaces produced with two 3D printing technologies. The test results showed a significant influence of the type of technology on the strength of the models built and on the quality of the technological surface layer. After the analysis of the parameters of the primary profile, roughness and waviness, it can be clearly stated that the quality of the technological surface layer is much better for the models made with the CFF technology compared to the FDM technology. Furthermore, the tensile strength of the models manufactured of carbon fiber-enriched material is much higher for samples made with CFF technology compared to FDM.

## 1. Introduction

Three-dimensional printing technologies have been known since the 1980s, when both the first program based on CAD modeling and the first 3D printers were invented. The most commonly used technologies are fused deposition modeling (FDM) and its equivalent fused filament fabrication (FFF). This is mainly due to the simplicity of the model building process, low cost of professional machines and cheap materials in terms of the price per kilogram. Furthermore, this technology is characterized by very large technological possibilities, such as the possibility of printing on already existing objects, which is particularly well described in publications [1,2,3,4]. Printing on existing objects also allows the construction of composite models with new innovative properties. Moreover, due to the large development of precision, FDM technology can be used in the future to build MEMS (microelectromechanical system) models [5]. Models produced with this technology can now be made using materials such as PLA (polylactic acid), ABS (acrylonitrile butadiene styrene), nylon (PA) and many different modifications of the materials mentioned. Thus, FDM technology has been widely deployed in many industrial sectors, due to not only the aforementioned advantages but also the ease to post process FDM-printed components, e.g., heat treatment, chemical treatment [6], machining [7,8], polishing, painting or coating. Considering the material modification for FDM, materials enriched with additives are becoming very popular, allowing for better mechanical properties, such as: additives that can reduce the hardness to even 30 on the Shore scale, flame retardant additives (aviation industry application), additives increasing mechanical properties, such as carbon fibers and glass fibers, and additives that enable the conduction of electricity and have magnetic properties. Specifically, the addition of glass and carbon fibers [9] seems very interesting, due to the fact that the process of building models using this material requires only a slightly higher temperature in the printing head extruder. As a matter of fact, glass and carbon fibers are very widely used in selective laser sintering (SLS) technology, where its addition improves the properties in such a way that it not only increases the strength of the manufactured models but also reduces the phenomenon of anisotropy of mechanical properties (especially in rheological tests) and accuracy, which is particularly visible in the production of thin-walled elements [10,11,12,13,14].

In comparison with tradition carbon fibers, which are a layer form, 3D printed carbon fibers produced with FDM technologies are solid and inherit all the aforementioned advantages due to the nature of 3D printing technologies and reinforcing ability of carbon fibers. Indeed, research related to the use of FDM/FFF technology, and PLA-based materials and carbon fibers additives, has been described in several research papers [15,16,17,18,19,20,21,22,23,24]. In particular, paper [16] presents an overview of the current research and the state of literature on PLA reinforced with glass and carbon fibers to improve the mechanical properties, with particular emphasis on thin-walled models and models with reduced mass. Besides, paper [17] presents a way forward to utilize FDM technology to manufacture continuous carbon fiber-reinforced thermoplastics samples. In the paper, several parameters, such as number of reinforced layers, material impact and interlayer gap, have been investigated and optimized using the response surface method. In the paper [18] the effects of stacking sequence of laminates and the effects of both short and continuous fibers’ contents, on the mechanical properties of laminated composites, were carefully analyzed by considering several layering configurations. Furthermore, in [19] the influence of sizing and printing process on interfacial performance and fracture patterns was studied systematically. The authors of the work [20] modified a FDM printer by implementing a new extruder that allows printing not only PLA but also PLA reinforced with carbon fiber. The subsequent tensile and bending tests showed that the addition of carbon fiber increases the strength of the models produced by over 30%, and in some cases, by over 50%. Additionally, pure PLA and PLA with 15% carbon fiber addition were studied in [21]. The printed samples were subjected to annealing followed by a microstructural study, and uniaxial tensile tests. In addition, the study carried out a dimensional analysis of the samples manufactured and indicated areas where shape errors occur. In [22], the surface characteristics, dimensional and shape accuracy, strength of components printed from pure PLA material (PLA and PLA 3D850) and HDPlas^®^ PLA material (PLA-graphene) are described in [23]. Compared to pure PLA, the graphene-enriched material has twice the tensile strength and a higher Young’s modulus. The samples were printed on different locations of the platform, and the surface analysis expressed by the spatial parameters of the Sa and Sz surfaces showed that the additives introduced into pure PLA increase the surface roughness, and the spatial parameters are of a slightly higher value.

The surface texture of 3D printed products should be analyzed based on the measured profile, not solely the roughness profile as in [25]. In particular, as for FDM technology, where the thickness of a single layer could be 0.2 mm, the surface quality should be examined for irregularities with longer wavelengths, i.e., surface waviness [4]. Additionally, the surface waviness is important, because it affects the vibration of mechanical components [26,27,28]. It should be mentioned that the influence of the 3D printing parameters on the quality of the surface layer may be more visible for the parameters of the primary profile and waviness than for the surface roughness [29]. Therefore, the surfaces texture should be analyzed in detail by analyzing the parameters of the roughness profile and waviness, and the primary profile. The surface roughness is a an important property of the fibers [30,31]. In the paper [30] a fractal model for capillary flow through a single tortuous capillary with roughened surfaces in fibrous porous media was presented. Moreover, the selection of the measurement method and filtration method is also important. For the analysis of surface irregularities, a Gaussian filter can be used; however, in recent years, a modern approach based on multiscale analysis has been increasingly applied for the evaluation of such surface textures [32,33,34].

Analyzing literature, it can be concluded that the introduction of additives in the form of carbon fibers to PLA-based materials is justified and significantly increases the strength properties. Different from the aforementioned papers, this manuscript not only investigates the strength, but also the surface roughness of the 3D printed carbon fibers from chopped carbon (PLA) produced by two common technologies being FDM and CFF. As a step further, the obtained results are compared with the ones from their extra reinforcement fibers (PA) counterparts, produced with CFF technology. Subsequently, it is possible to point out which technology is better in terms of the mechanical properties and the surface roughness of the components that it produces. This would provide readers better view on the current technical solutions to 3D print carbon fibers, how to assess their properties, which would open up more opportunities for developing applications with thin-walled and reduced mass, such as those used in the aerospace and automotive industries.

## 2. Materials and Methods

### 2.1. Method

The test samples were designed in accordance with ASTM D 638, type V using the CAD system. During the research, two 3D printing technologies (FDM and CFF) were used to build physical models of samples. The Makerbot Replicator 5th Gen printer was used for the FDM technology, and the Markforged X7 machines for the CFF technology. Measurements of the geometrical structure of the surface were then carried out on the flat surface of the samples using a TOPO 01P L120 stylus profilometer. After completing the measurements and determining the parameters of the surface profile, waviness and roughness, the samples were subjected to uniaxial tensile strength tests using a testing machine (Instron, 5980 Series) [35].

#### 2.1.1. FDM Technology

FDM technology is one of the most common 3D printing technologies. In this method, the plastic-based material in the printing head is heated to a temperature slightly lower than the melting point and is then pressed through the nozzle and distributed in the place where the layer of the cross-section of the model is currently built. In this technology, we used materials based on plastics such as PLA, ABS, NYLONS and materials enriched with additives, such as the carbon fiber used in the work. Makerbot Replicator 5th Gen printers were used to build sample models using FDM technology. The FDM printer is characterized by a working chamber with dimensions of 25.2 length × 19.9 width × 15.0 height cm^3^. The machine is equipped with an extruder that allows the material to be heated to a temperature of 250 °C. In addition, the machine enables the production of models with a variable diameter of the printing nozzle from 0.1 to 0.4 mm.

#### 2.1.2. CFF Technology

Continuous filament fabrication (CFF) is an improved technology patented by the commercial company Markforged. In principle, the technology is based on FDM. Structurally, the printer includes a printhead with two independent extrusion nozzles. One nozzle is intended for printing plastic filaments and the other for printing reinforcing fiber [36]. These two nozzles do not work simultaneously, but the first one prints the plastic filament. It is then stopped, the second nozzle prints the continuous fiber on the previous plastic layer, thereby integrating the layer, and can continue printing the first nozzle. The resulting part is then a composite of two materials. CFF can print functional components by continuous fiber reinforcement (fiberglass, carbon fiber, Kevlar or HSHT fiber), which is inserted into a polymer matrix based on PA6 (white nylon) or into a modified filament known under the trade name Onyx, which contains a PA6 polymer matrix with chopped carbon fibers (Onyx, Onyx FR—fire resistance and Onyx ESD—electrostatic discharge). The Onyx material itself has high flexural strength (according to [37] 71 MPa), and in combination with carbon fiber, it has even better properties (according to [37] flexural strength 540 MPa).

A variant of CFF is a technology-based on a very similar mechanism, sometimes referred to as continuous fiber reinforced thermoplastic composites (CFRTPC). This technology also operates on an FDM basis, where instead of two printing nozzles, only one is used, in which the supplied plastic is mixed with the reinforcement filament and fed into the same nozzle but with a separate inlet. After heating the nozzle, the matrix with the reinforcing fiber is infused, and the layer is applied to the printing bed [38].

### 2.2. Materials

Four (4) types of samples manufactured of different materials were selected for the tests, where ten samples were made for each type (40 in total). The first type was a material utilizing FDM technology—PLA with 20% carbon fiber (Carbonfil—infill density 95%), which in the tests, next to the sample number, was marked with the letter—T. The second type of sample was manufactured utilizing CFF technology, a material based on PA (ONYX—95% infill), which, during the tests, next to the sample number, was marked with the letter—O. The third type of sample was models made of polyamide—PA (ONYX), but with a specified 37% infill, marked with the symbol—ON. The last type of sample was models marked with the letter—C, made of Onyx + carbon fibers.

Carbonfil is PLA enriched with the addition of 20% carbon fiber. This material is in the form of a rod with a diameter of 1.75 mm. The PLA material, with the structural formula, is shown in Figure 1 [39,40]. The manufacturer’s recommended extruder temperature during printing is in the range of 230–265 °C (we used 250 °C).

All CFF samples in this study were manufactured using a desktop 3D printer (Markforged^®^ Mark X7, Watertown, Massachusetts, USA) with a preset layer height of 0.100–0.125 mm, using a continuous CF/PA6 fiber. The material was supplied by Markforged. A total of three groups of samples were produced utilizing CFF technology for the tensile test. The first group was made purely of Onyx material with a full solid infill. The second group was made of 37% Onyx infill, which is the default settings, and the last group was reinforced in certain layers with carbon fibers (more information about settings is in Section 2.3).

When using the reinforcing fiber, Onyx is used as the matrix material. This Onyx filament is a mixture of nylon plastic material and chopped carbon fibers. A continuous carbon fiber is used to reinforce the fibers. Onyx fiber has a diameter of 1.75 mm, while carbon fiber has a diameter of 0.33 mm. The reinforcing carbon fiber is not really a single fiber but a tangle of fibers embedded in a polyamide (PA) matrix. The paper [41] reported that these fibers are not evenly distributed and tend to agglomerate in groups, leaving gaps between them. The size of the individual fiber is then, in the cross-section, 6.9 ± 0.7 μm. Since [41] used the same filament manufacturer for the tests as in our study, it can be assumed that the agglomeration in the reinforcing fiber will be the same.

Selected properties of Carbonfil, Onyx and carbon fibers (for CFF) are presented in Table 1.

### 2.3. Preparation of Samples

Test samples were designed according to Figure 2 using SolidWorks (Dassault Systems SolidWorks Corp., Waltham, Massachusetts) and then approximated with triangles to create STL files. In order to accurately reproduce the 3D model of the samples, the saving of STL files took into account two tolerance parameters: linear tolerance—0.01 mm and angle tolerance—1°. The 3D sample model saved as an stl file has been approximated by 652 triangles and is shown in Figure 3. The sample models on the building platform with the marking of the metrologically measured surface is shown in Figure 4.

Using FDM technology, the samples were made of PLA material with carbon fibers with the following technological parameters: layer thickness—0.2 mm, extruder temperature 250 °C and infill material—95%.

To prepare the samples by CFF technology, it is first necessary to create a 3D model and then convert it to a stereolithography (STL) file, as with the models for FDM. A file prepared in this way can then be uploaded to the Eiger cloud slicer software, supplied exclusively by Markforged. Eiger software not only controls the deployment of reinforcing fibers, but it is also possible to tune the printing parameters (fill pattern, fill density, roof and floor layers and wall layer), fiber type, fiber fill type, number of fiber layers, fiber orientation, fiber rings, etc. All these parameters affect the resulting mechanical properties of the part. Eiger allows you to either send the build directly to the printer or export the build.

For experimental testing, three sets of samples in CFF technology were prepared, each with ten pieces. The first set, marked “O” (Figure 5a), had the main print set at infill 100%, thus full solid. Samples were made only from Onyx with a setting of two wall layers, 0.100 mm layer height and an Onyx material consumption of 1.56 cm^3^. The second set, labeled “ON” (Figure 5b), had a major infill printing parameter of 37% of pure Onyx with a setting of pattern triangular, two wall layers, 0.100 mm layer height and material consumption of 1.09 cm^3^. The last set of samples was reinforced with carbon fiber, marked “C”. The fiber placement is shown in Figure 5c, with the fiber fill type Isotropic chosen due to the distribution of the fiber throughout the sample. The fiber was laid in layers 5–8 and then 19–22, so there are a total of 8 layers of carbon fiber reinforced in the sample. In the remaining layers, the infill of the plastic material was set to 37% and, as in the previous samples of two wall layers, due to the reinforcing fiber, the layer height was set to 0.125 mm. This setting respects the recommended setting by Eiger. The consumption of Onyx material was 1.56 cm^3^, and carbon continuous fiber 0.26 cm^3^. The selected technological properties are presented in Table 2.

### 2.4. Measurement Methods

The samples obtained as a result of the above-mentioned technologies were subjected to quality assessment of the obtained surface texture. The surface topography analysis was carried out on the basis of the analysis of the primary profile, waviness and roughness of the profiles. Measurements were made with the use of a TOPO 01P stylus profilometer on the measuring section, which was selected, taking into account the sample size, which was equal to 8.8 mm. This size was chosen due to the sample size, the filtering process (Gaussian filter and edge effects) and the capabilities of the measuring machine. The TOPO 01P profilometer is a professional measuring system designed for measuring surface roughness by the stylus method on flat cylindrical external and internal surfaces. TOPO 01P is a stationary device with high accuracy and a wide measuring range. Due to the wide measuring possibilities using all additional possibilities of the device, TOPO 01P can be used both for measuring selected parameters of surface roughness of workpieces, and for comprehensive experimental tests in the field of metrology of the geometric structure of the surface. The measurement was carried out by moving the measuring head, which carried out its movement perpendicular to the printing track. The measurement allowed for the creation of a profile with the horizontal sampling density Δx = 1 µm. In order to ensure the randomness of the distribution of irregularities on the surface of the sample, measurements were made three times on each of the ten sample surfaces for each material.

Tensile tests were performed using an Instron Universal Testing Machine (Model 3382). Five specimens were tested for each composition. The displacement rate for the tensile tests was kept to 1 mm/min. Tensile test properties were carried out according to ASTM D-638 type V, with a specimen thickness of 3 mm.

## 3. Results

### 3.1. Surface Texture

Tests carried out on samples made with the use of additive technologies utilizing selected machines were analyzed in multiple ways. The surface texture consists of the form, waviness and roughness; therefore, it was decided to consider the primary profile, and its components. Analysis of the surface quality was based on the analysis of selected profile parameters. For this purpose, the roughness, waviness and primary profile of samples were analyzed. The evaluation was carried out quantitatively for selected parameters from each group of parameters, i.e., peaks and valleys in the height direction, average amplitude in the height direction, average characteristics in the height direction, hybrid and horizontal direction. From the first group of parameters, the parameters determining the arithmetic means and geometric deviations of the profile, skewness, kurtosis and the profile height were selected, which are described by parameters for the primary profile Pa, Pq, Psk, Pku and Pt, respectively. Moreover, it was decided that in order to make a comprehensive assessment of surface texture, it is necessary to analyze parameters from particular groups. The following parameters were selected: mean width of the primary profile, mean height of the primary profile, root mean square slope of the primary profile and the number of local peaks of the primary profile, which are described for the primary profile as PSm, Pc, Pdq and PPc. For the waviness and roughness of the profile, analogous markings (the letters W and R, respectively) were used. The results of the analysis for the selected parameters of the primary profile, waviness and roughness are shown in Figure 6.

Analyzing the results obtained for parameters, it can be concluded that there is some analogy for the tested materials. For the samples made with CFF technology, the obtained values of the parameters determining the resulting irregularities differ from the values determined for the samples made with FDM technology. This is especially visible for the parameters defining the arithmetic mean height and the root mean square deviation. On the basis of Figure 6, it can be observed that both the obtained values and the determined uncertainty for the probability equal to 0.95 was much lower for the samples marked as C, O and ON. This proves the much better quality of the surface texture of manufactured samples. For the samples, it can also be noticed that for this technology on the resulting surfaces and roughness, i.e., the resulting short-term irregularities, is responsible for the main component of irregularities, while FDM technology results in the formation of long-term profile irregularities. Moreover, it should be noted that there are similar values of kurtosis parameters, which were obtained for all materials. It can be concluded that the distribution of peaks and valleys on all samples was similar. For the other analyzed parameters shown in Figure 6, it should be noted that there was no significant difference between the obtained parameter values. For both technologies, the contribution of roughness and waviness to the profile irregularities was similar. However, it should be noted that the obtained values of selected parameters were higher for FDM technology than the values calculated for the surface of samples made with CFF technology.

ANOVA analysis with post-hoc Tukey’s test was used to determine the significance of the different parameter values in the examined coefficients. For this reason, the *p*-value was calculated. Table 3 and Table 4 present the calculated values in terms of the assessed profile parameters.

Analyzing the results of the one-way ANOVA statistics, it should be stated that for the parameters determining skewness and kurtosis, no significant difference was noted between the sample surfaces with each of the four analyzed methods. The calculated values were lower than the test value of the statistics, which was also shown in the table, where the critical value sufficient to find the similarity was considered as *p* > 0.05. For the other parameters, the obtained values of Tukey’s test showed that the sample marked as T is significantly different from the rest of the samples.

Table 4 presents the results of the analysis for the remaining parameters. On the basis of the presented data, it should be stated that when significant differences between the values for the surface were observed, this was for the sample marked as T. For samples made with the CFF method, significant differences in parameters occurred sporadically.

### 3.2. Tensile Test

The results of the tensile measurements of samples reinforced with carbon fiber are shown in Figure 7, Figure 8, Figure 9 and Figure 10, where Figure 7 presents a summary of the tensile results of the samples, which very well illustrates the impact of both the technology used and the type of so-called “infill” material.

The stress–strain curve is shown in Figure 7 above. It can be seen that the highest peak of the curve was observed for the sample C (CFF), followed by the sample T (FDM) with 20% of carbon fiber. Sample C showed a remarkable tensile strength compared to samples T, O and ON. The curves showed that samples ON and O exhibit ductile and low strength characteristics in terms of strain values.

The average tensile strength results are shown in Figure 8. The lowest tensile strength of 29 MPa was observed for samples made from a polyamide base utilizing CFF with printing parameter of 37% infill, which are also labeled as “ON”. For samples marked with “O” (polyamide-based, CFF), the tensile strength increased to 42%. For sample marked with “T” (PLA with carbon fiber in FDM), the strength for the material was not much higher than for the PA-based material (ON and O) utilizing CFF technology. For samples reinforced with carbon fiber—C, a significant increase of more than 250% was observed compare to ON, O and T. From the results obtained, it was found that better tensile strength values were achieved with an increased infill density in PA-based material utilizing CFF, and PLA samples reinforced with carbon fiber. “ON”, “O” and “T” have almost the same category of strength. “C” has a significant improvement of tensile strength. The lowest strength was found in the case of “ON” with 9.6 MPa. Samples reinforced with carbon fiber “C” improved by 234% compared to ON.

The variation of Young’s modulus is presented in Figure 9. The lowest Young’s modulus was for sample CFF—ON (134 MPa). The Young’s modulus increased to 81% in the case of sample O (CFF). Further enhancements in Young’s modulus were also observed for samples T (FDM) and C (CFF), where the values increased 816% and 967%, respectively. The bundles of carbon fibers are responsible for withstanding the forces applied to the samples, with a high elastic modulus and a limited maximum elongation [42].

The variation of tensile strain is shown in Figure 10. Samples with labels T and C exhibit low tensile strain, which means the samples were brittle. However, samples reinforced with carbon fibers (C) showed characteristics of high stiffness compared with the samples reinforced with carbon fibers (T). Besides this, the sample reinforced with carbon fibers (C) has a slightly higher strain than the sample with carbon fiber T, though with better tensile strength. Sample T has the lowest strain and low strength. The strain value for the sample marked ON recorded a tensile strain value of 55%, whilst the sample marked O had the highest strain of about 76%. In general, samples ON and O showed ductile properties, low strength and low stiffness.

The results of the tensile tests revealed that the type of technology had a substantial impact on the strength of the samples built and the quality of the technological surface layer. It can be concluded that the tensile properties of the technological surface layer are superior for samples created with CFF technology compared to samples produced with the FDM method. The polymeric matrix that encapsulates the continuous fiber ensures strong adhesion between the reinforcement filament and the previous layer or surrounding material. As a result, this latest advancement of CFF technology producing polymer composites with superior tensile properties compared to FDM technology.

Figure 11 shows microscopic SEM images taken with a magnification of 150×, 200×, 500× and 4000×.

As can be seen in Figure 11 above, photographs “c” and “d” clearly show the carbon fibers of the reinforced materials. However, in the case of samples T and photograph “c”, made with magnification 150×, the fibers were evenly distributed over the entire fracture surface of the samples, due to the fact that with FDM technology, the PLA-based material was mixed with carbon fibers and constitutes an integral whole. In the case of “c” samples, which were made utilizing CFF technology, the carbon fibers were present only in a certain part, which corresponded to the layer made of just carbon as a whole. In photo “d”, taken with a magnification of 150× and 200×, the regions (regions of layers) with the accumulation of carbon fibers were clearly visible.

## 4. Discussion

The quantitative analysis of the test results shows that the use of material—C (ONYX + carbon fiber) allows for the production of models with the highest strength, even over 100% greater than material—T (Carbonfil—PLA + carbon fibers), and it is almost 3 times more durable than pure ONYX with 100% and 37% filling. Moreover, it can be seen that both carbon fiber reinforced materials have a low elongation of less than 1 mm compared to PA-based Onyx 100% and Onyx 37%, where elongation was 5 mm. In the case of Young’s module analysis, it can be said that carbon fiber reinforced materials are characterized by a very high value of over 1200 for T material utilizing FDM technology and over 1400 for C material utilizing CFF technology. The much higher mechanical properties of material C (CFF) result from the characteristic way of building the interior of the models. In this technology, there is a high concentration of fibers pressed together and wrapped together like a rope, which guarantees high tensile strength. It seems that it would be reasonable to carry out compression and tensile tests taking into account stress relaxation tests during continuous loads in time.

The analysis of the parameters of the primary profile, roughness and waviness allows one to conclude that the arithmetical mean high, i.e., Pa, Wa and Ra, were obtained for samples made with FDM technology, and for these samples, the scatter of results was also the largest. The roughness parameters for all types of samples range from Ra = 5 to 8 µm. In the case of the waviness analysis and the Wa parameter, it can be seen that for the sample models made with CFF technology, the value of this parameter in all variants was similar and was below 4 µm. In the case of FDM technology, it was clearly visible that the Wa parameter took a value above 13 µm, which was over 3 times higher. Similarly, for the other analyzed parameters, it was observed that the values calculated for the surface of the samples made with FDM technology were higher both in terms of the obtained values and their scatter. The variation of irregularities in relation to the surface analyzed using a one-way ANOVA showed that the obtained values of roughness parameters that describe the surfaces were significantly different from the values obtained for CFF technology. For the samples made with this method, the textures of the samples were similar to each other. While evaluating the surface texture, one should also pay attention to its type. For CFF technology, the main component of errors was included in short-term irregularities, i.e., roughness, while for FDM, in the waviness of the surface.

The analysis of SEM microscopic images taken with the given magnification levels of 150×, 200×, 500× and 4000× clearly shows the carbon fibers, which were particularly visible in the case of samples made with FDM technology, the cross-section of which was full, without any visible voids inside. In all photos, especially at a lower magnification, you can see the layered structure characteristic of 3D printing. In the case of samples—T, the carbon fibers were visible on the entire surface, and in the case of samples C, where carbon was present only in selected layers, their concentration in selected parts of the cross-section can be seen. In all types of samples, it can be seen from the rupture that the PLA-based and PA-based materials were characterized by quite high porosity in all variants of their execution.

A completely different way of building composite models in the case of both 3D printing technologies means that both the metrological and strength results were completely different. It seems that the method of laying the layers of material based on polyamide PA, and then laying specific layers of carbon fiber material, allows one to provide much greater strength of the models produced, which is shown in Figure 7 and Figure 8. Moreover, microscopic observation of SEM indicates a much greater accumulation of fibers carbon in a few selected places in the case of C (CFF) samples, and not spreading them as uniformly as it is in the case of T (FDM) samples. Although carbon fibers look similar in the case of higher magnifications, their greater accumulation provided by the CFF technology, as indicated by the measurement results, ensures a much better end result in the form of higher strength parameters, both in relation to pure PLA and Carbonfil and Onyx.

## 5. Conclusions

The following conclusions can be drawn from the analysis of the presented results of research of the surface texture and tensile tests of samples manufactured with two technologies (FDM and CFF) and materials based on PLA and PA and reinforced with carbon fibers:The addition of carbon fibers greatly affected the strength of the manufactured models—increasing both the tensile strength and the modulus of elasticity. It seems that models manufactured in CFF technology, where only selected layers consist of carbon material, show the greatest strength.The analysis of SEM microscopic photos shows that in the case of samples based on PA (O and ON samples), both types of infill material had high porosity, and there were voids in the variant of 37% filling, which negatively affected the strength of the models. Carbon fibers were uniformly visible in the case of samples manufactured with FDM technology and in a layered manner for CFF technology, which, as can be seen from the strength, shows the advantage of the second variant of placing additives in the form of carbon fibers.The presented studies of the surface texture have shown that there is a dependence between the obtained surface texture of the samples and the applied technology. Depending on the method used, a different distribution of surface irregularities was obtained, in which, depending on the technology, waviness or roughness was the dominant component.One-way ANOVA analysis showed only sporadically significant differences in the values of parameters between samples made by CFF technology. On the other hand, these differences were observed to a larger extent when comparing the values of parameters of waviness, roughness and primary profile for samples made with the use of the two analyzed technologies.Summarizing the results of the research in which both FDM and CFF 3D printing technologies were compared, many interesting conclusions showed that the addition of carbon fiber alone is not a factor determining high mechanical properties, the method of fiber supply depending on the 3D printing method is also important. It seems that the use of CFF technology and a material based on carbon fiber reinforced polyamide for the construction of thin-walled models is the right solution, which, as shown by measurements of the surface structure, ensures better roughness.

## Figures and Tables

**Figure 1 polymers-13-01671-f001:**
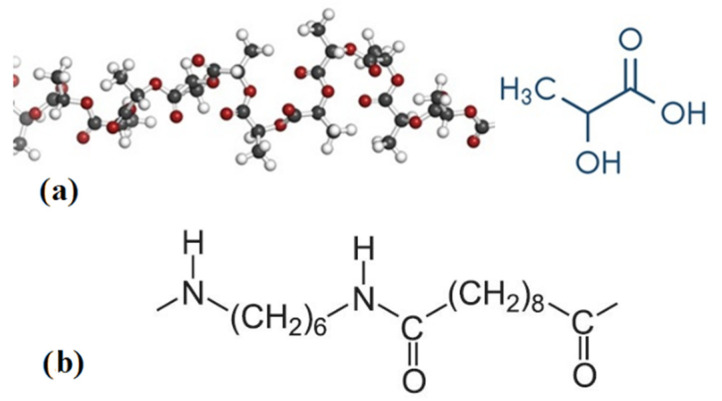
Structural model of: (**a**) PLA; (**b**) PA.

**Figure 2 polymers-13-01671-f002:**
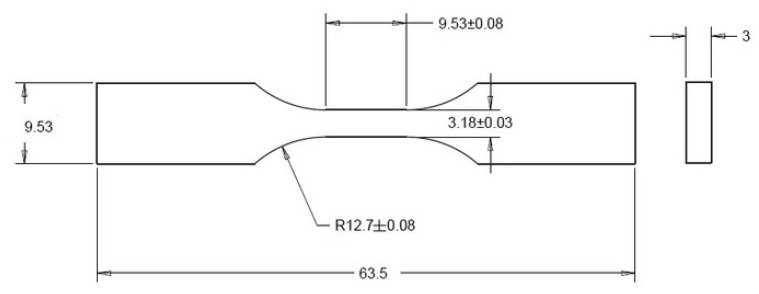
Sample dimensions.

**Figure 3 polymers-13-01671-f003:**
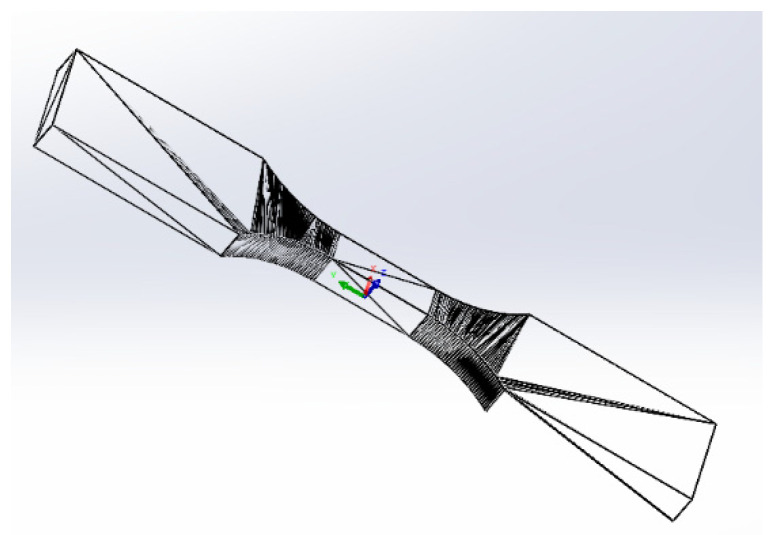
STL sample model.

**Figure 4 polymers-13-01671-f004:**
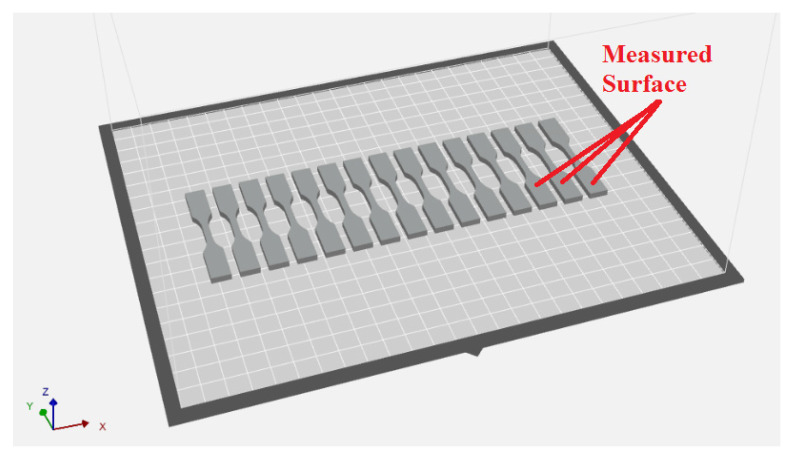
Three-dimensional model on the building tray.

**Figure 5 polymers-13-01671-f005:**
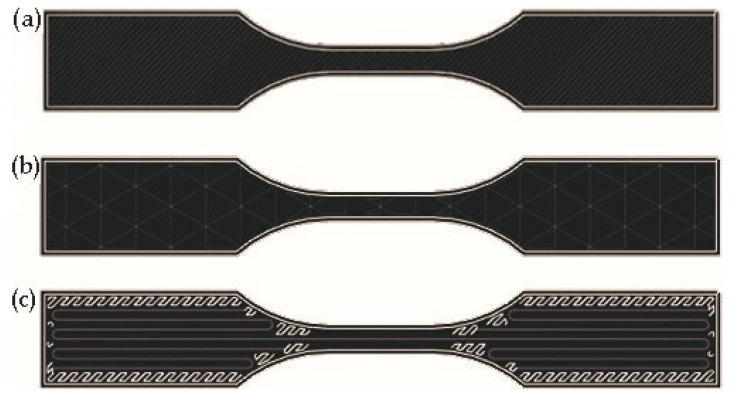
Sample preparation by CFF technology—sample marked “O” (**a**); sample marked “ON” (**b**); sample marked “C” (**c**)—blue lines indicate reinforcing carbon fiber, white lines indicate Onyx filament.

**Figure 6 polymers-13-01671-f006:**
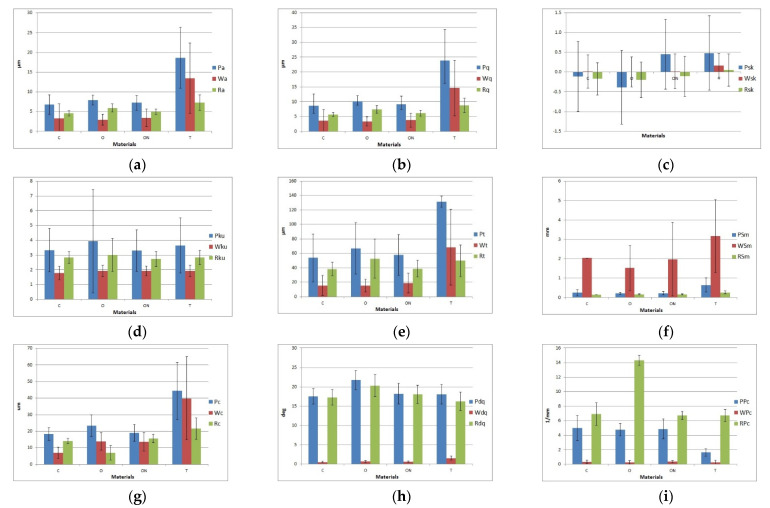
Surface parameters: (**a**) Pa/Wa/Ra, (**b**) Pq/Wq/Rq, (**c**) Psk/Wsk/Rsk, (**d**) Pku/Wku/Rku, (**e**) Pt/Wt/Rt, (**f**) PSm/WSm/RSm, (**g**) Pc/Wc/Rc, (**h**) Pdq/Wdq/Rdq and (**i**) PPc/WPc/RPc.

**Figure 7 polymers-13-01671-f007:**
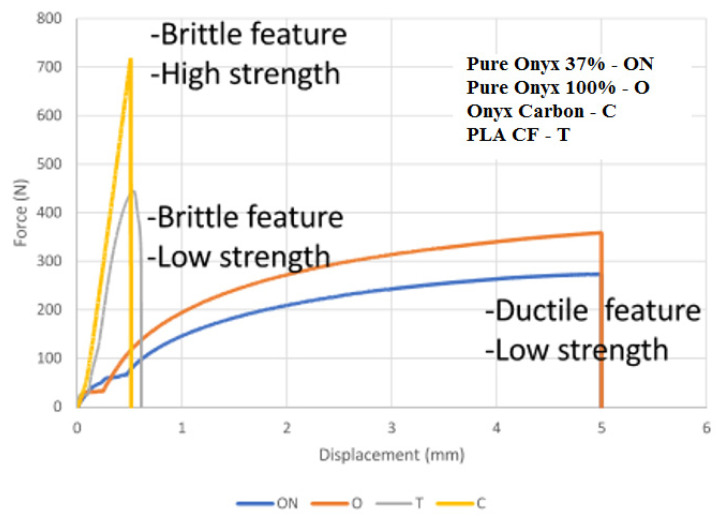
Tensile test results.

**Figure 8 polymers-13-01671-f008:**
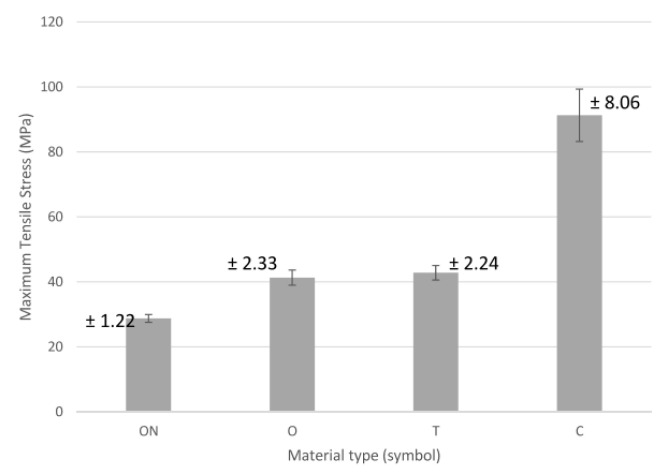
Average tensile strength.

**Figure 9 polymers-13-01671-f009:**
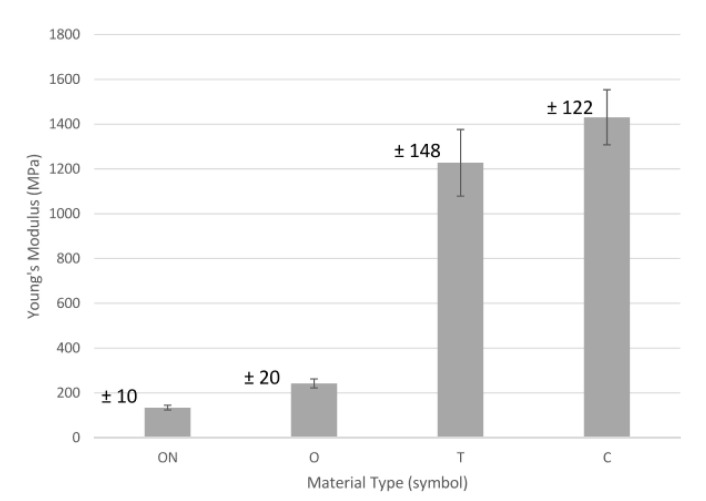
Young’s modulus.

**Figure 10 polymers-13-01671-f010:**
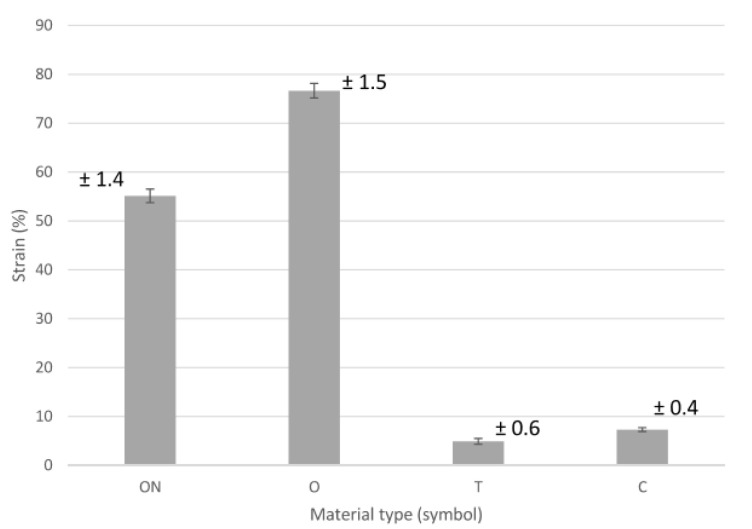
Tensile strain.

**Figure 11 polymers-13-01671-f011:**
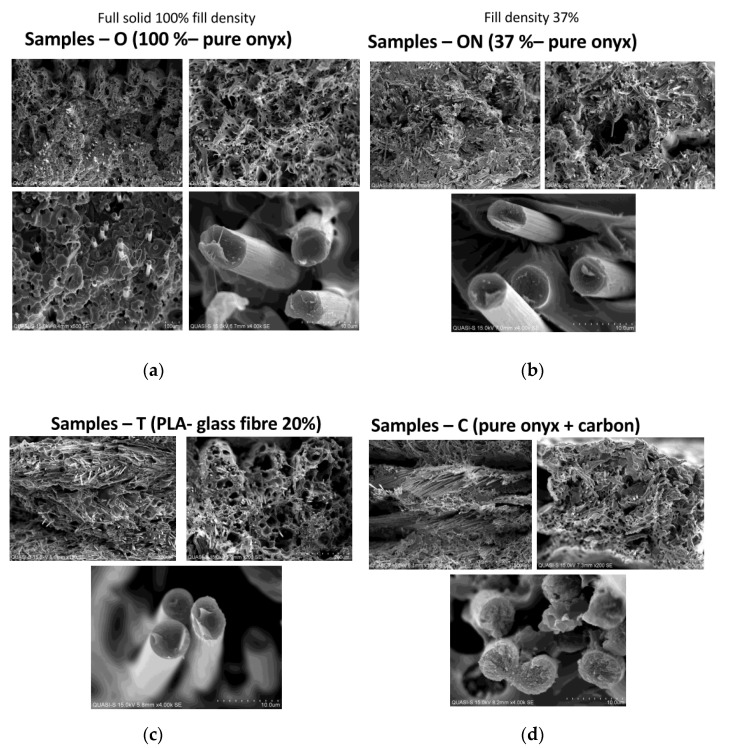
SEM microscopy results.

**Table 1 polymers-13-01671-t001:** Properties of used materials.

Properties	Carbonfil	Onyx	Carbon Fibers (CFF)
Specific gravity	1.19 g/cm^3^	1.2 g/cm^3^	1.4 g/cm^3^
Impact strength	7.9 KJ/m^2^ (ASTM D792)	0.33 KJ/m (D256-10 A)	0.96 KJ/m (D256-10 A)
Tensile modulus	3800 MPa (ASTM D256)	2400 MPa (D638)	60,000 MPa (D638)
Elongation at break	8% (ISO 527)	25% (D638)	1.5% (D3039)
Print temperature	±230–265 °C	±270 °C	±250 °C

**Table 2 polymers-13-01671-t002:** Settings of tested samples.

Sample	Layer Height (mm)	Infill (-)	Fill Density (%)	Reinforcement (-)	No. of Layers (-)	Filament Consumption (cm^3^)
O	0.100	Full solid	100	-	32	1.56
ON	0.100	Triangular	37	-	32	1.09
C	0.125	Triangular	37	Carbon Fibers	26/8	1.56/0.26
T	0.200	Full solid	95	Carbon Fibers	16	1.93

**Table 3 polymers-13-01671-t003:** ANOVA analysis—part one.

	Pa	Pq	Psk	Pku	Pt	Wa	Wq	Wsk	Wku	Wt	Ra	Rq	Rsk	Rku	Rt
C:O	0.56052	0.629928	0.562303	0.64953	0.397417	0.989091	0.995307	0.99962	0.293084	0.999997	0.000142	8.88 × 10^−5^	0.987628	0.630119	0.009705
C:ON	0.952145	0.975447	0.917981	0.999837	0.96512	0.99952	0.998654	0.99986	0.935582	0.958012	0.554604	0.599038	0.887035	0.928208	0.999427
C:T	3.77 × 10^−9^	3.77 × 10^−9^	0.949736	0.999811	5.54 × 10^−9^	4.90 × 10^−9^	4.26 × 10^−9^	0.379502	0.93208	1.54 × 10^−8^	4.70 × 10^−8^	3.99 × 10^−7^	0.120838	0.474314	0.048985
O:ON	0.859765	0.860392	0.225224	0.602362	0.680432	0.974073	0.979474	0.999102	0.099345	0.95306	0.006442	0.003454	0.720634	0.284806	0.013146
O:T	3.77 × 10^−9^	3.78 × 10^−9^	0.268985	0.698237	1.92 × 10^−7^	4.24 × 10^−9^	4.02 × 10^−9^	0.326845	0.096993	1.48 × 10^−8^	0.052979	0.301727	0.060152	0.054372	0.913548
ON:T	3.77 × 10^−9^	3.78 × 10^−9^	0.999575	0.998603	1.06 × 10^−8^	5.31 × 10^−9^	4.52 × 10^−9^	0.397974	0.999999	5.51 × 10^−8^	2.36 × 10^−6^	1.80 × 10^−5^	0.413538	0.832496	0.063714

**Table 4 polymers-13-01671-t004:** ANOVA analysis—part two.

	PSm	Pc	Pdq	PPc	WSm	Wc	Wdq	WPc	RSm	Rc	Rdq	RPc
C:O	0.997059	0.393189	0.000227	0.927642	0.999805	0.716727	0.8039596	0.668703	0.779801	0.009563	0.000209	9.70 × 10^−1^
C:ON	0.999327	0.99661	0.817928	0.986484	0.970443	0.73294	0.932891	0.997436	0.85164	0.663348	0.731507	0.938505
C:T	8.43 × 10^−6^	1.77 × 10^−6^	0.889543	5.58 × 10^−7^	5.18 × 10^−1^	6.61 × 10^−3^	6.46 × 10^−7^	0.869892	6.48 × 10^−9^	2.48 × 10^−4^	5.59 × 10^−1^	3.76 × 10^−7^
O:ON	0.999822	0.508183	0.001267	0.992553	0.952718	0.999991	0.989322	0.555353	0.998882	0.091186	0.001614	0.7419495
O:T	6.03 × 10^−6^	2.68 × 10^−5^	0.000921	1.40 × 10^−6^	4.72 × 10^−1^	5.48 × 10^−2^	2.60 × 10^−6^	0.980451	1.14 × 10^−8^	2.98 × 10^−1^	1.83 × 10^−5^	2.01 × 10^−7^
ON:T	6.87 × 10^−6^	2.46 × 10^−6^	0.998507	9.21 × 10^−7^	7.74 × 10^−1^	5.19 × 10^−2^	1.59 × 10^−6^	0.776959	1.02 × 10^−8^	2.44 × 10^−3^	1.25 × 10^−1^	8.76 × 10^−7^

## Data Availability

The data presented in this study are available on request from the corresponding author.

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
