# Peer review of "Quality of Surface Texture and Mechanical Properties of PLA and PA-Based Material Reinforced with Carbon Fibers Manufactured by FDM and CFF 3D Printing Technologies"

_polymers, 2021, doi:10.3390/polym13111671_

Round 1

Reviewer 1 Report

3D printing technologies have been known since the 1980s, when both the first program based on CAD modeling and the first 3D printers were invented. Over the course of 40 years, there has been a dynamic development of 3D printing technology. New machines implementing layered technologies were created, and the technological possibilities and accuracy of the existing systems were increased. Currently, 3D printers enable the production of not only prototypes but also fully functional models with real application in many industries, such as heavy industry, casting, injection molding, aviation, automotive, medical, precision and military industries. In this article, the paper presents the results of mechanical tests of models manufactured with two 3D printing technologies, FDM and CFF. Both technologies use PLA or PA-based materials reinforced with carbon fibers. The work includes both uniaxial tensile tests of the tested materials and metrological measurements of surfaces produced with two 3D printing technologies. Although the topic in this work was interesting, the presentation in this manuscript was very poor. This manuscript should be rejected for published in Polymers. However, if the authors are willing to make the substantial revisions according to my comments, I would be glad to re-review this manuscript. Here are my detailed comments:

  1. The introduction should be reconstructed to present a coherent literature review. It may help the authors by answering the following questions: Why are these works relevant? Which specific problems were addressed? How are the previous results related with the latest work? What are the outstanding, unresolved, research issues? Answering the questions leads to the novelty of the proposed work naturally.
  2. Materials and Methods part, Although the results look “making sense”, the current form reads like a simple lab report. The authors should dig deeper in the results by presenting some in-depth discussion.
  3. It is suggested to discuss what the main advantages the 3D printing technologies have.
  4. The test results showed a significant influence of the type of technology on the strength of the models built and on the quality of the technological surface layer. After the analysis of the parameters of the primary profile, roughness and waviness, it can be clearly stated that the quality of the technological surface layer is much better for the models made with the CFF technology compared to the FDM technology. Furthermore, the tensile strength of the models madnufactured of carbon fiber-enriched material is much higher for samples made with CFF technology compared to FDM. The authors should give some explanation on above conclusions.
  5. In this article, the surface analysis expressed by the spatial parameters of the Sa and Sz surfaces showed that the additives introduced into pure PLA increase the surface roughness, and the spatial parameters are of a slightly higher value. The TOPO 01P profilometer is a professional measuring system designed for measuring surface roughness by the stylus method on flat cylindrical external and internal surfaces. The surface roughness is an important property of fibers. And several investigators have studied the effect of surface roughness on mechanical properties of fibers, see [A fractal model for capillary flow through a single tortuous capillary with roughened surfaces in fibrous porous media, Fractals, 2021, 29(1):2150017; Fractals, 2019, 27(7): 1950116 ]. Authors should introduce some related knowledge to readers. I think this is essential to keep the interest of the reader.
  6. Please, expand the conclusions in relation to the specific goals and the future work.

Author Response

First at all, thank you for your time spent by reviewing our paper, it certainly improved the quality of the article. We have done our best with the revisions. We hope the manuscript will now fully meet the journal requirements. Please find below the comments to your review. Also you can see what was changed in „Track changes“ in word file.

1. The introduction should be reconstructed to present a coherent literature review. It may help the authors by answering the following questions: Why are these works relevant? Which specific problems were addressed? How are the previous results related with the latest work? What are the outstanding, unresolved, research issues? Answering the questions leads to the novelty of the proposed work naturally.

Thank you very much for your notes about Introduction. Whole section of Introduction was reconstructed and now the paragraphs has clearer and understable flow, which is easier to follow. The questions were answered in last paragraph (lines 116-126)

2. Materials and Methods part, Although the results look “making sense”, the current form reads like a simple lab report. The authors should dig deeper in the results by presenting some in-depth discussion.

Thanks for your suggestion. We improved the article so that we completed the results section (lines 331-334, 367-373), discussion (lines 396-400, 424-433) and the conclusion.

3. It is suggested to discuss what the main advantages the 3D printing technologies have.

Discussion of advantages can be found on lines 51-61 also in lines 68-73 we tried to focus more on advantages of 3D print with reinforcement which is the main topic of paper.

4. The test results showed a significant influence of the type of technology on the strength of the models built and on the quality of the technological surface layer. After the analysis of the parameters of the primary profile, roughness and waviness, it can be clearly stated that the quality of the technological surface layer is much better for the models made with the CFF technology compared to the FDM technology. Furthermore, the tensile strength of the models madnufactured of carbon fiber-enriched material is much higher for samples made with CFF technology compared to FDM. The authors should give some explanation on above conclusions.

Thank you for the feedback. An explanation text has been added (lines 367-373)

5. In this article, the surface analysis expressed by the spatial parameters of the Sa and Sz surfaces showed that the additives introduced into pure PLA increase the surface roughness, and the spatial parameters are of a slightly higher value. The TOPO 01P profilometer is a professional measuring system designed for measuring surface roughness by the stylus method on flat cylindrical external and internal surfaces. The surface roughness is an important property of fibers. And several investigators have studied the effect of surface roughness on mechanical properties of fibers, see [A fractal model for capillary flow through a single tortuous capillary with roughened surfaces in fibrous porous media, Fractals, 2021, 29(1):2150017; Fractals, 2019, 27(7): 1950116 ]. Authors should introduce some related knowledge to readers. I think this is essential to keep the interest of the reader.

References related with the effect of surface roughness on mechanical properties of fibers have been added into manuscript (it can be found in references 31 and 32)

6. Please, expand the conclusions in relation to the specific goals and the future work.

Thank you for this suggestion. We supplemented part of the conclusion with a short summary in relation to the main research goal in point 5 in conclusion (lines 457-463)

Reviewer 2 Report

In this manuscript, the authors studied the mechanical properties of models which were manufactured with PLA or PA-based materials reinforced with carbon fibers using FDM and CFF printing. When simply searching the keywords online, similar studies have already done. For example, composites Part B: Engineering, 124, 88-100; Composites Part B: Engineering, 175, 107147. Composite Structures, 207, 232-239; Composites Part A: Applied Science and Manufacturing, 114, 368-376. So this study has limited impact. Here are some major concerns.

  1. The authors should rewrite the introduction. The background of 3D printing is unless. It looks like a review paper. The authors failed to discuss the innovation of the current study.
  2. It is useless to write the structural formula of PLA. Where is PA?
  3. The model is too simple. The authors should design more models to show the printability and printing features.
  4. In Figure 6, the statistical analysis should be performed. The error bar is too large. There is no significant difference between the different samples.
  5. The authors failed to discuss the advantages and rationale of FDM and CFF printing.

Author Response

First at all, thank you for your time spent by reviewing our paper, it certainly improved the quality of the article. We have done our best with the revisions. We hope the manuscript will now fully meet the journal requirements. Please find below the comments to your review. Also you can see what was changed in „Track changes“ in word file.

  1. The authors should rewrite the introduction. The background of 3D printing is unless. It looks like a review paper. The authors failed to discuss the innovation of the current study.

Thank you very much for your notes about Introduction. Whole section of Introduction was reconstructed and reduced, now the paragraphs has clearer and understable flow, which is easier to follow and doesn’t look like review paper. The discuss about innovation now can be found on last paragraph od Introduction (lines 116-126).

  1. It is useless to write the structural formula of PLA. Where is PA?

The formula for PA was added in the figures a and in addition we add 1a for PLA and 1b for polyamide.

  1. The model is too simple. The authors should design more models to show the printability and printing features.

Thank you for this suggestion. Each type of sample was made in the amount of 10 pieces to take into account the repeatability of the test results (40 samples in total). It seems that in the future it would be appropriate to conduct more extensive research with more samples e.g. 30 for each parameters. Also the shape of model was according to standard.

  1. In Figure 6, the statistical analysis should be performed. The error bar is too large. There is no significant difference between the different samples.

The statistical analysis of the measurement results of the surface geometric structure was carried out on the basis of the results obtained from the TOPO 01 profilometer. Large error bars are caused by a large variation of parameter values caused by irregularities of the tested samples.

  1. The authors failed to discuss the advantages and rationale of FDM and CFF printing.

Discussion of advantages can be found on lines 51-61 also in lines 68-73 we tried to focus more on advantages of 3D print with reinforcement which is the main topic of paper. The whole section of Introduction is reconstructed, also the aforementioned discuss.

Round 2

Reviewer 1 Report

In Ref. 31, “Fractals. 29 (2021)” should be corrected as “Fractals 2021, 29, 2150017”; In Ref. 32, “Fractals. (2019)” should be corrected as “Fractals 2019, 27, 1950116”

Reviewer 2 Report

The authors have answered my comments. Although I don't think the quality of this study is good enough to publish in this journal, the editor gave a revision decision. So I do respect the editors' opinion.